# Low-Frequency Air–Bone Gap and Pulsatile Tinnitus Due to a Dural Arteriovenous Fistula: Considerations upon Possible Pathomechanisms and Literature Review

Andrea Tozzi [1], Andrea Castellucci [2,*], Giuseppe Ferrulli [1], Salvatore Martellucci [3], Pasquale Malara [4], Cristina Brandolini [5], Enrico Armato [6] and Angelo Ghidini [2]

[1] Otorhinolaryngology-Head and Neck Surgery Department, University Hospital of Modena, 41125 Modena, Italy; andreatozzi29@gmail.com (A.T.); dottorgiuseppeferrulli@gmail.com (G.F.)

[2] ENT Unit, Department of Surgery, Azienda USL—IRCCS di Reggio Emilia, 42123 Reggio Emilia, Italy; angelo.ghidini@ausl.re.it

[3] ENT Unit, Santa Maria Goretti Hospital, Azienda USL di Latina, 04100 Latina, Italy; dott.martellucci@gmail.com

[4] Audiology & Vestibology Service, Centromedico, 6500 Bellinzona, Switzerland; pasmalara@gmail.com

[5] Otorhinolaryngology and Audiology Unit, IRCCS Azienda Ospedaliero-Universitaria di Bologna, Policlinico S. Orsola-Malpighi, 40138 Bologna, Italy; cristina.brandolini@aosp.bo.it

[6] Faculty of Medicine, University of Lorraine, 54000 Vandoeuvre-lès-Nancy, France; armato.otovest@gmail.com

\* Correspondence: andrea.castellucci@ausl.re.it; Tel.: +39-0522-296273; Fax: +39-0522-295839

**Abstract:** Low-frequency air–bone gap (ABG) associated with pulsatile tinnitus (PT) and normal impedance audiometry represents a common finding in patients with third window syndromes. Other inner disorders, including Meniere's disease (MD), perilymphatic fistula and intralabyrinthine schwannoma, might sometimes result in a similar scenario. On the other hand, PT is frequently associated with dural arteriovenous fistula (DAVF), while conductive hearing loss (CHL) is extremely rare in this clinical setting. A 47-year-old patient was referred to our center with progressive left-sided PT alongside ipsilateral fullness and hearing loss. She also experienced headache and dizziness. Otoscopy and video-oculographic examination were unremarkable. Conversely, a detailed instrumental audio-vestibular assessment revealed low-frequency CHL with normal impedance audiometry, slight left-sided caloric weakness, slightly impaired vestibular-evoked myogenic potentials on the left and normal results on the video-head impulse test, consistent with an MD-like instrumental profile. Gadolinium-enhanced brain MRI revealed an early enhancement of the left transverse sinus, consistent with a left DAVF between the left occipital artery and the transverse sinus, which was then confirmed by angiography. A trans-arterial embolization with Onyx glue was performed, resulting in a complete recession of the symptoms. Post-operatively, the low-frequency ABG disappeared, supporting the possible role of venous intracranial hypertension and abnormal pressure of inner ear fluids in the onset of symptoms and offering new insights into the pathomechanism of inner ear CHL.

**Keywords:** conductive hearing loss; air–bone gap; pulsatile tinnitus; dural arteriovenous fistula; inner ear fluids; venous intracranial hypertension; pseudo-conductive hearing loss; Meniere's disease

## 1. Introduction

Tinnitus and conductive hearing loss (CHL) are common symptoms usually managed in everyday clinical practice for otolaryngologists. However, this association can offer a diagnostic challenge if not combined with middle ear abnormalities. Tinnitus may be defined as a conscious awareness of a sound in the absence of an external auditory stimulus. If the sound can also be heard by the examiner, it is defined "objective"; when it is synchronous with the heartbeat, it is called pulsatile tinnitus (PT) and could represent a sign of a vascular abnormality [1]. Advances in neuroimaging and endovascular treatment have led to the increased detection of vascular causes for PT, which could be distinguishable

based on the involved vascular structure. Idiopathic intracranial hypertension, dural venous sinus abnormalities (e.g., stenosis, diverticula and bony dehiscences), emissary vein anomalies and jugular vein alteration (like a high jugular bulb) are among the most common venous causes of PT. On the other hand, arterial causes of PT include carotid/vertebral artery stenosis, fibromuscular dysplasia, aneurysm, abnormal arteries in the middle ear (e.g., aberrant internal carotid artery and persistent stapedial artery) and arteriovenous malformations (AVMs) like dural arteriovenous fistula (DAVF) [2].

DAVFs represent pathological shunts between dural arteries and dural venous sinuses, meningeal veins or cortical veins, accounting for 10–15% of intracranial AVMs [3]. Most DAVFs leading to the onset of symptoms in adulthood are located in the transverse and sigmoid sinuses [4]. Although its etiopathogenesis is still poorly understood, trauma, craniotomies and dural venous thrombosis seem to play a key role in the development of DAVFs [5]. Besides PT, DAVFs can result in an array of symptoms depending on the location and include bruit, headache, visus modification, mental status alterations, seizure, myelopathies, cranial nerve palsies and motor/sensory deficits. Approximately 20–30% of DAVFs present with intracranial hemorrhage [6]. Current classification systems of DAVFs, including the most widespread classifications according to Cognard and Borden, are based on the different patterns of venous drainage, which is known to predict the clinical evolution (Table 1) [7,8]. The lack of cortical venous drainage (Borden Type-I, Cognard type-I and IIa) represents a favorable feature and is associated with a benign natural history. While digital subtraction angiography (DSA) represents the diagnostic gold standard for the detection and evaluation of DAVFs, computed tomography and magnetic resonance angiography (CTA and MRA, respectively) can be considered useful tools for screening patients, highlighting red flags (i.e., dilatated vessels, venous pouches, vascular enhancement and sign of intracranial venous hypertension) and planning the most appropriate treatment [9]. Generally, in cases of asymptomatic benign DAVFs, conservative management is preferred, while treatment is recommended for both symptomatic and aggressive fistulas. Various therapeutic options are now available, including endovascular embolization, open surgery and radiosurgery. While the former is considered the first-line treatment for DAVFs using both trans-arterial and trans-venous approaches, with different embolic agents (Cyanoacrylic glue, Onyx, and the most recent Squid and PHIL agents), surgery and stereotactic radiosurgery remain an alternative option, particularly when an endovascular approach is unsuccessful or dangerous [10–12].

On the other hand, CHL associated with PT and normal impedance audiometry represents a key finding for the diagnosis of inner ear disorders, leading to a third mobile window mechanism (TMWM). In fact, according to the most widely accepted theories, labyrinthine capsule abnormalities, including canal dehiscence, enlarged vestibular aqueduct (EVA) and an array of other bony defects exposing the inner ear sensors to the action of the surrounding structures, account for a low-impedance pathway for sounds and pressure stimuli resulting in a TMWM [13–16]. In these cases, the travelling wave in response to low-frequency air-conducted (AC) sounds is shunted away from the cochlea to the vestibular compartment, leading to an increased auditory threshold for low-frequency AC sounds and, simultaneously, an increased sensitivity to bone-conducted (BC) stimuli—this is why it is also called "pseudo-CHL" [13,17]. TMWM should also result in abnormal vestibular-evoked myogenic potentials (VEMPs) to both AC and BC sounds, presenting with enhanced amplitudes and lowered thresholds [18,19]. On the other hand, in these disorders, the pulse-synchronous activity of the vascular structures surrounding the otic capsule is directly conveyed to the inner ear fluids and to the cochlear partition, resulting in a typical PT that might be occasionally objectified through a pulsatile wave pattern on impedance audiometry [20–24]. Therefore, low-frequency air–bone gap (ABG) and PT without middle ear abnormalities, together with enhanced VEMPs, should address the diagnosis toward a TMWM. Actually, additional inner ear disorders including perilymphatic fistula, early-stage Meniere's disease (MD) and intralabyrinthine schwannoma might result in CHL with no abnormalities on impedance audiometry. While the former might lead to

a CHL through a mechanism similar to a TMWM [25], endolymphatic hydrops (EH) and intralabirinthine schwannomas might account for cochlear hydrodynamic changes interfering with the endolymphatic travelling wave to AC sounds, resulting in CHL variably associated to sensorineural hearing loss (SNHL) [13,26–32]. Nevertheless, canal and/or otolith function is usually impaired in these cases, and instrumental vestibular tests often result in abnormal findings. In particular, MD has been demonstrated to generate a wide range of abnormalities for saccular and utricular function (as tested by cervical and ocular VEMPs, respectively) [33,34], and a typical dissociation between horizontal semicircular canal (HSC) function in the low- and high-frequency domains, as tested by the bithermal caloric test (BCT) and the video-head impulse test (vHIT), respectively. Namely, while the vestibulo-ocular reflex (VOR) of the HSC in usually spared on the vHIT, even in the late stage of the disease, caloric response is frequently impaired on the same side on the BCT [35,36]. Different theories have been proposed for this functional dissociation, including a herniation of the membranous labyrinth into the HSC preventing cupular displacements to caloric stimuli and a dissipation of endolymphatic waves in response to caloric irrigations due to a distension of the membranous ducts on a hydropic basis [37,38]

**Table 1.** Classifications of dural arteriovenous fistulas.

| Borden Classification | | Cognard Classification | |
|---|---|---|---|
| Type I | Anterograde drainage into the dural sinus/meningeal vein | Type I | Anterograde drainage into venous sinus |
| Type II | Anterograde drainage into dural sinus and retrograde drainage into cortical veins | Type II:<br>• IIA<br>• IIB<br>• IIA+IIB | Drainage into main sinus with reflux into secondary sinus<br>Drainage into main sinus with reflux into cortical vein<br>Drainage into main sinus with reflux into secondary sinus and cortical veins |
| Type III | Isolated retrograde drainage:<br>• Drains directly into cortical veins<br>• Trapped segment of sinus with reflux into cortical veins<br>• Venous varix/dural lake with reflux into cortical veins | Type III | Direct cortical venous drainage without ectasia |
| | | Type IV | Direct cortical venous drainage with venous ectasia |
| | | Type V | Drainage into the spinal perimedullary veins |

In this study, we describe the clinical and instrumental findings of a patient with a DAVF between the left occipital artery and the transverse sinus, presenting with low-frequency CHL and PT that receded after endovascular treatment. Interestingly, preoperative instrumental findings were consistent with left MD. A similar case was previously described by Cassandro et al., and our case represents the second description of this unusual combination in the literature [39]. We also review the current published literature reporting the association between DAVF and HL, and discuss the possible relationship between cerebrospinal fluid (CSF) pressure, vascular intracranial anomalies and inner ear fluid dynamics.

## 2. Case Presentation

A 47-year-old female was referred to us with a two-month history of progressive left-sided PT, aural fullness and slight ipsilateral HL. She noted that tinnitus became incapacitating during physical efforts. She also experienced headache and dizziness, and denied vertigo attacks or a history of head trauma. Her body mass index was 21. She was then submitted to a complete audio-vestibular assessment. Otoscopic examination revealed normal tympanic membranes. A pulsatile bruit could be palpably appreciated over her left

mastoid tip and could be almost totally abolished with a soft compression of the same area. Pure tone audiometry revealed a normal hearing threshold on the right side and a low-frequency ABG in her left ear. Type-A tympanograms with normal acoustic reflexes could be registered (Figure 1). Neither spontaneous nor positional nystagmus was detected at the video-oculographic examination, while a slight right-beating nystagmus could be elicited after head shaking and skull vibrations. The BCT showed a slight unilateral weakness on the left side, whereas the vHIT resulted in normal function for all semicircular canals (Figure 2a,b). Both cervical VEMPs to AC sounds and ocular VEMPs to BC stimuli were asymmetrical with right-sided prevalence, according to normative values for asymmetry ratio [34] (Figure 2c). Due to the aforementioned instrumental findings, MD was postulated first, despite the lack of acute vertigo spells. A temporal bone high-resolution CT (HRCT) scan completed by reformatted images along the plane of the vertical semicircular canals excluded labyrinthine dehiscences or jugular bulb abnormalities, while a gadolinium-enhanced angio-MRI of the brain revealed an early enhancement of the left transverse sinus using time-resolved imaging of contrast kinetics sequences (Figure 3a). Neither signs of empty sella nor distention of the perioptic subarachnoid sheaths were found. Carotid artery ultrasounds confirmed dilatation of the left occipital artery with a low-resistance and high-velocity flow, raising the hypothesis of a left DAVF between the left occipital artery and the dural transverse sinus. The patient was then scheduled for an intracranial DSA. At the late arterial phase after injecting the left external carotid artery, a wide DAVF supplied by branches of the left occipital artery with an early drainage into the transverse sinus (type-I according to Congard classification) could be clearly detected (Figure 3b). The DAVF was concurrently treated with an endovascular embolization of the feeding branches using Onyx glue. A post-operative DSA showed a restoration of the physiological intracranial circulation (Figure 4a), and the patient reported an immediate receding of both PT and fullness together with hearing recovery on the left side. Pure tone audiometry documented a complete closure of the pre-operative ABG (Figure 4b). The patient was then discharged and followed-up for 2 years. Neither PT relapse nor hearing deterioration were reported.

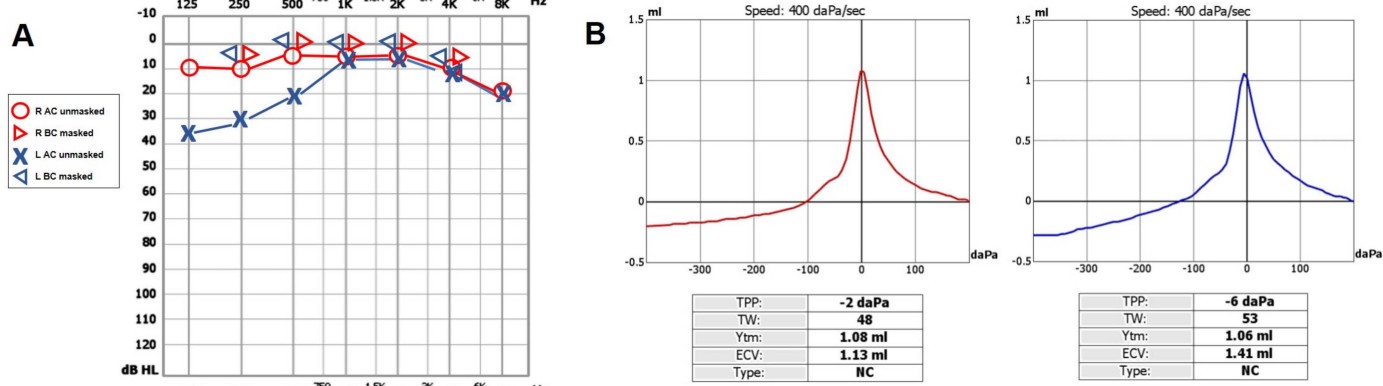

**Figure 1.** Pre-operative cochlear assessment. (**A**) Pure tone audiometry conducted in a soundproof room using standard clinical procedures showing normal hearing threshold on the right- and left-sided low-frequency CHL. (**B**) Tympanometry exhibiting type-A profiles on both sides. AC: air conducted, BC: bone conducted, CHL: conductive hearing loss, L: left, R: right.

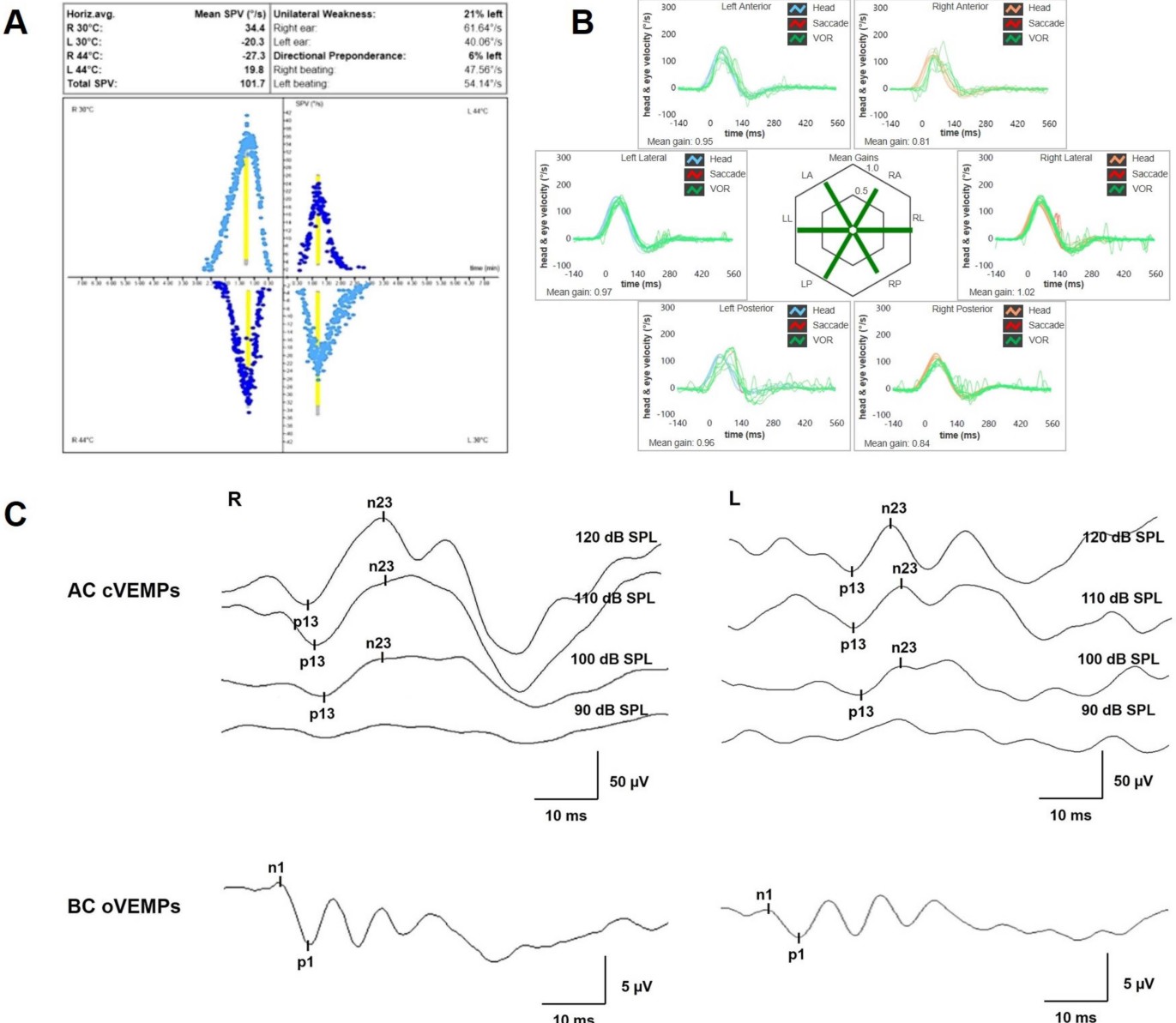

**Figure 2.** Pre-operative vestibular assessment. (**A**) BCT was assessed using sequential water irrigations at 30/44 °C according to the Fitzgerald–Hallpike method. Responses were evaluated using a VOG system (model VN415/VO425 Firewire, Interacoustics, Denmark), and UW was calculated using Jongkees' formula. In this case, a left-sided UW of 21% was detected. (**B**) vHIT performed using the ICS Impulse device (GN Otometrics, Denmark) assessing all six semicircular canal VORs in horizontal, LARP and RALP planes. Mean value of VOR gain (eye velocity/head velocity) is reported for each canal. Gains are considered normal if >0.8 for lateral canals and >0.7 for vertical canals. A normal VOR gain for all semicircular canals with neither covert nor overt saccades was registered. (**C**) Cervical VEMPs to AC sounds (above) and ocular VEMPs to BC stimuli (below) were recorded using a 2-channel evoked potential acquisition system (Epic plus, Labat, Italy). Cervical VEMPs to AC sounds were recorded delivering tone bursts (frequency, 500 Hz; duration, 8 ms; stimulation rate, 5 Hz) via headphones. For BC ocular VEMPs, 500 Hz tone bursts (8 ms, 1.0 V, 0.6 A) were delivered to the midline (Fz) by a hand-held minishaker with an attached Perspex rod (type 4810, Bruel & Kjaer P/L, Denmark). Vibratory stimulation was varied in intensity and amplification through a power amplifier (type 2718, Bruel & Kjaer P/L, Denmark). VEMPs testing revealed slightly asymmetrical amplitudes for both cervical VEMPs (R 95.5 µV and L 42.8 µV at 120 dB SPL stimuli, AR: 38%) and ocular VEMPs (R 5.8 µV and L 2.7 µV for 120 dB SPL sounds, AR: 36%). AC: air-conducted, AR: asymmetry

ratio, BC: bone-conducted, BCT: bithermal caloric test, L: left, LA: left anterior, LL: left lateral, LP: left posterior, R: right, RA: right anterior, RL: right lateral, RP: right posterior, UW: unilateral weakness, VEMPs: vestibular-evoked myogenic potentials, vHIT: video-head impulse test, VOG: video-oculography, VOR: vestibulo-ocular reflex.

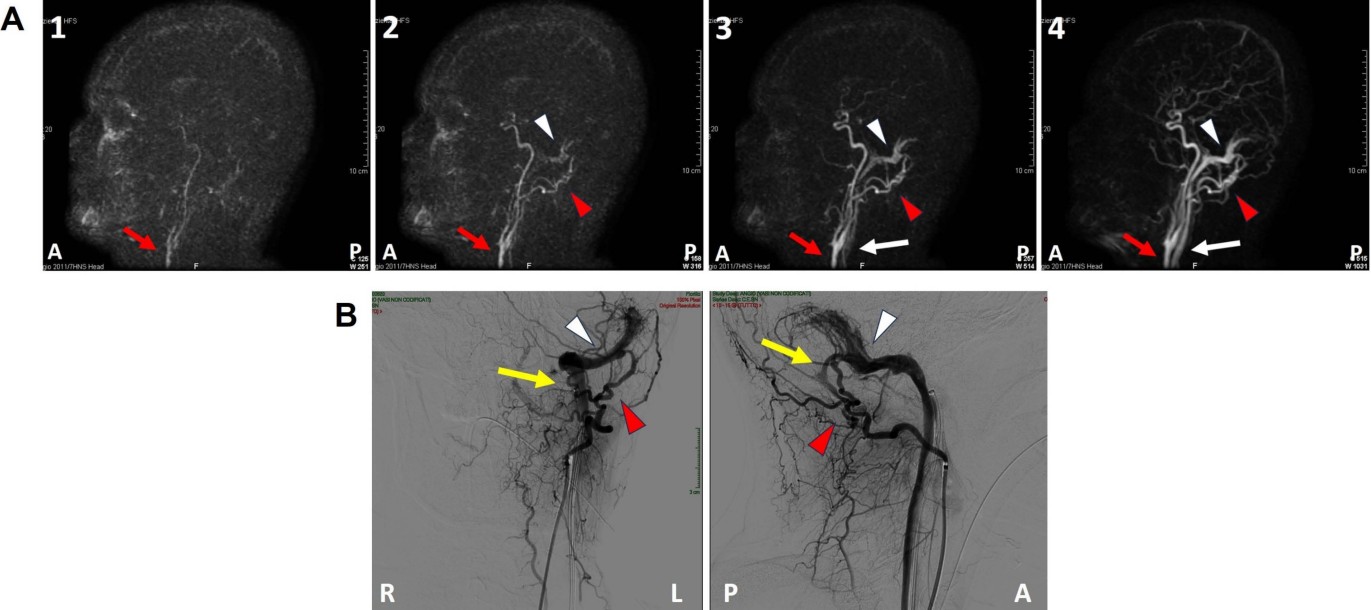

**Figure 3.** Pre-operative imaging. (**A**(**1–4**)) Brain angio-MRI sequences on the sagittal plane using the TRICKS protocol showing early enhancement of the left dural transverse sinus (white arrowhead). The red arrow indicates the carotid artery trunk, the red arrowhead the left occipital artery and the white arrow the internal jugular vein. (**B**) Coronal (right side) and sagittal (left side) projections of the intracranial DSA showing a type-I DAVF (yellow arrow) between the left occipital artery (red arrowhead) and the dural transverse sinus (white arrowhead). A: anterior, DSA: digital subtraction angiography, L: left, MRI: magnetic resonance imaging, P: posterior, R: right, TRICKS: time-resolved imaging of contrast kinetics.

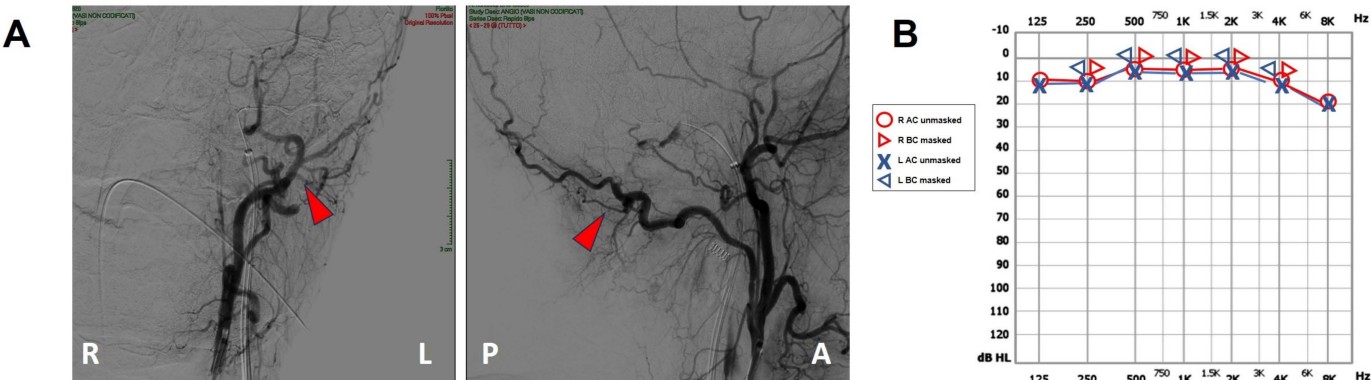

**Figure 4.** Post-operative findings. (**A**) Coronal (right side) and sagittal (left side) projections of the intracranial DSA showing the sole left occipital artery (red arrowhead). Neither DAVF nor transverse sinus can be detected. (**B**) Pure tone audiometry showing normalization of the AC threshold on the left side with closure of the ABG. A: anterior, ABG: air–bone gap, AC: air-conducted, BC: bone-conducted, DSA: digital subtraction angiography, L: left, MRI: magnetic resonance imaging, P: posterior, R: right.

## 3. Discussion

While PT is widely considered a typical manifestation of DAVFs [40], the association between AVM and other otologic manifestations such as vertigo, otalgia and chronic otitis

media with effusion represent anecdotal findings [41–43]. Similarly, AVMs presenting with HL have rarely been reported in the literature. To our knowledge, only seven papers reporting the association between DAVFs and HL have been published so far (Table 2). Weider et al. first described two original cases with AVM causing low-frequency HL, either SNHL or CHL, which improved after treatment. Nevertheless, the patient with CHL did not develop a closure of the ABG, but both AC and BC thresholds improved accordingly with the assumption of selective low-frequency masking by AVM pulsations and with residual tinnitus after surgery [44]. In two additional papers, HL was ascribed to other underlying pathological conditions, including superficial siderosis and vestibular schwannoma, which likely resulted in structural damage of the inner ear sensors that did not recover post-operatively [45,46]. Similarly, a possible cochlear nerve/internal auditory artery compression by the DAVF nidus was assumed by Kim et al. to be the most likely pathomechanism behind SNHL, as reported in a patient [47]. The type of HL was not specified in two additional cases [48,49]. Finally, Cassandro et al. reported a complete receding of CHL and PT after endovascular embolization of a DAVF, similarly to our patient. Authors supposed a subclinical intracranial hypertension with abnormal venous drainage and, in turn, an excess of endolymphatic volume, resulting in an injury in the microcirculation of the stria vascularis [39,50].

**Table 2.** Reported cases of DAVF associated with hearing loss. CHL: conductive hearing loss, DAVF: dural arteriovenous fistula, SNHL: sensorineural hearing loss.

| Author | Year | Study | DAVF Type | Hearing Loss | Treatment | Hearing Recovery |
|---|---|---|---|---|---|---|
| Weider D.J. [44] | 1990 | Case Report | / | SNHL | embolization | Yes |
|  |  |  | / | CHL | surgical resection | Yes |
| Kim M.S. et al. [47] | 2002 | Case Report | / | SNHL | embolization | No |
| Cassandro E. et al. [39] | 2015 | Case Report | II Cognard | CHL | Onyx glue embolization | Yes |
| Baum G.R. et al. [45] | 2016 | Case Report | III Borden | SNHL | Onyx glue embolization | No |
| Gioppo A. et al. [48] | 2017 | Case Report | III Cognard | / | squid embolization | / |
| Kritikos M.E. et al. [46] | 2018 | Case Report | IIa + b Cognard | SNHL | surgical resection + Onyx glue embolization | No |
| Peto I. et al. [49] | 2019 | Case Report | IV Cognard | / | surgical resection | Yes |

Before providing a possible explanation for the symptoms and signs depicted in our patient, a brief recall on the relationships between intracranial vascular structures, CSF and inner ear fluids (perilymph and endolymph) should be presented. The premise is the Monro–Kellie principle, which states that the intracranial compartment is an incompressible structure and that the overall volume of its contents remains constant. A DAVF represents an alteration in venous intracranial circulation: the lack of high resistances in the arteriolar bed and the presence of a low-resistance nidus result in a high blood-flow rate and hypotension in the feeding artery combined with intranidal and draining vein hypertension. An impairment of cerebral venous outflow can produce intracranial hypertension, mimicking benign intracranial hypertension [51]. Several mechanisms have been involved in this interesting relationship. Kühner postulated that the pathogenesis of intracranial hypertension in DAVFs could be related to increased dural sinus pressure, resulting in a diminution of the CSF absorption, which, in turn, could lead to increased intracranial hypertension [52]. More recently, the "hydraulic hypothesis" introduced by Rossitti addresses the attention to the microscopic pathophysiology, highlighting the transmission of high draining-vein pressure to the subarachnoid veins and, consequently, to the whole CSF space, which can induce a general subarachnoid vein compression even far away from the AVM draining vein [53].

It should be also mentioned that intracranial and intralabyrinthine fluids are deeply connected: while the perilymph communicates with the intracranial fluids through the

cochlear aqueduct, the endolymph is connected with the intracranial fluids via the endolymphatic sac. The balance between perilymph and endolymph is maintained by the Reissner's membrane and other labyrinthine membranes. The cochlear aqueduct seems to act as a low-pass filter, preventing the spreading of the variation in intracranial hypertension to the inner ear. This function could lose effectiveness due to interindividual age-related decline [54].

According to the aforementioned mechanisms, we can reasonably assume that, in our case, a type-I DAVF resulted in general intracranial venous hypertension due to abnormal venous drainage. This mechanism might lead to a non-structural and reversible clinical manifestation, even far from the AVM nidus, generating subclinical intracranial hypertension, as reported by Hurst et al. [55]. Nevertheless, neither clinical nor radiological signs of intracranial hypertension were found, except for the headache, though an extensive neuro-ophthalmologic evaluation was not carried out. The abnormal high CSF pressure could have been transmitted to the inner ear following the aforementioned pathways, generating perilymphatic hydrops and EH [56], which could likely result in HL. Similar events have been reported in cases of increased CSF pressure due to stenosis of the aqueduct of Sylvius [57,58]. The increased perilymphatic and endolymphatic pressure, together with possible saccular dilatation, could likely result in a dampened stapedial mobility, thus increasing AC sounds impedance and, in turn, resulting in a low-frequency CHL. Several authors assumed the same mechanisms to explain the ABG in patients with MD [13,26–28,30,32]. Notably, Eggermont and Schmidt described a case with unilateral MD in which, during symptomatic exacerbation of the affected ear, an ABG appeared in the opposite side [59]. The idea of increased inner ear fluid pressure with possible saccule dilatation and impairment of stapedial footplate mobility has been supported recently by studies on MD patients with 3T MRI, showing that the average ABG was significantly higher in the ears exhibiting EH adjacent to the stapedial footplate and, more generally, that ABG significantly correlated with the degree of EH [30,60]. Accordingly, EH signs were detected in our patient through an extensive vestibular assessment: in particular, the dissociation between BCT and vHIT results, and the reduced VEMPs to AC and BC sounds, directed our first diagnosis toward MD [33–38]. Symptoms receding and ABG closure following the endovascular treatment of the DAVF, likely due to the restoration of physiological values of intracranial pressure resulting in the recovery of normal inner ear micromechanics, strongly support this theory. Moreover, similar alterations in inner ear fluids dynamics have also been implied in the genesis of CHL in EVA, where signs of EH have been detected through imaging [61]. In fact, patients with EVA frequently present with mixed HL, including both low-frequency ABG and high-frequency SNHL, which can either fluctuate or progress to severe or profound HL, preventing the detection of the ABG [62]. Besides the aforementioned TMWM, according to which EVA would act as a low-impedance pathway for AC sounds, as proposed by Merchant et al. [13,63], increased perilymphatic and endolymphatic pressure could result in reduced stapes mobility, leading to CHL, as originally hypothesized by Valvassori and Clemis [64]. The latter theory could also explain the higher incidence of gusher in EVA patients undergoing cochlear implants compared to other patients with profound SNHL [65,66]. Additionally, recent studies have demonstrated how a subgroup of MD patients might be related to altered intracranial venous drainage, particularly with chronic cerebrospinal venous insufficiency, as their symptoms improved following endovascular treatment [67].

Even though the data obtained from pre-operative instrumental assessment and the restoration of symptoms and signs after the treatment of the DAVF strongly support our theories offering new insights into the relationships between intracranial and intralabyrinthine fluids, the present report presents some limitations. First, CSF pressure was not measured, nor was the presence of papilledema checked. Also, delayed MRI acquisitions following contrast injection to objectively detect signs of EH were not obtained, as 3T MRI was not available in our institution. Moreover, despite the patient herein reported exhibiting an MD-like instrumental profile, she neither reported acute vertigo spells nor had fluctu-

ating auditory symptoms, resulting in an incomplete clinical picture for MD diagnostic criteria [68]. Additionally, neither frequency tuning for VEMPs nor electrocochleography were performed to provide additional clues for EH. Similarly, a comprehensive vestibular assessment was not pursued in the post-operative days to ascertain the normalization of pre-operative abnormalities. Further studies are needed to support our hypothesis and better clarify the relationship between DAVFs and inner ear symptoms.

## 4. Conclusions

When low-frequency CHL and PT occur together with normal middle ear status, additional pathomechanisms other than TMWM should be searched for, including increased pressure of intralabyrinthine fluids. DAVFs can likely result in increased intracranial venous pressure, generating subclinical CSF hypertension, which, in turn, might lead to perilymphatic hydrops and EH. Further studies are needed to support these interesting insights on the relationships between the intracranial venous system and the homeostasis of intralabyrinthine fluids.

**Author Contributions:** Conceptualization, A.C.; methodology, A.T., A.C., S.M. and P.M.; investigation and data collection, A.C. and C.B.; writing—original draft preparation, A.T. and G.F.; writing—review and editing, A.C., S.M. and P.M.; supervision, E.A. and A.G. All authors have read and agreed to the published version of the manuscript.

**Funding:** This research received no external funding.

**Institutional Review Board Statement:** This research was conducted in accordance with ethical principles, including the World Medical Association Declaration of Helsinki (2002). The local ethical committee of the Area Vasta Emilia Nord does not perform formal ethical assessments for case reports.

**Informed Consent Statement:** Written informed consent was obtained from the patient to publish this paper.

**Data Availability Statement:** Not Applicable.

**Acknowledgments:** In memory of Giovanni Carlo Modugno.

**Conflicts of Interest:** The authors declare no conflict of interest.

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
