# Peer review of "Low-Frequency Air–Bone Gap and Pulsatile Tinnitus Due to a Dural Arteriovenous Fistula: Considerations upon Possible Pathomechanisms and Literature Review"

_audiolres, doi:10.3390/audiolres13060073_

Round 1

Reviewer 1 Report

Comments and Suggestions for Authors

The manuscript titled with “Low-frequency Air-bone Gap and Pulsatile Tinnitus due to Dural Arteriovenous Fistula: Considerations upon Possible 3 Pathomechanisms and Literature Review” reported an interesting case of pulsatile tinnitus due to dural arteriovenous fistula. The authors discussed on the possible mechanism. It is the second case of the similar manifestation in the literature. It is relevant to the clinic to report the second case. However, the manuscript needs revision.

1.     The sentence on page 7 L227-228 that “Finally, Cassandro et al. reported a complete receding 227 of CHL and PT after an endovascular embolization of a DAVF, similarly to our patient” should gone to introduction instead of discussion.

2.     On page 8 L263-264: “This mechanism might lead to non-structural and reversible 263 clinical manifestation even far from the AVM nidus, generating a subclinical intracranial 264 hypertension, as reported by Hurst et al. [55]”. Did the authors measure CSF pressure in the present case?

3.     On page 8 L271-273: “The increased perilymphatic and endolymphatic pressure, together with a possible saccular dilatation, could likely result in a dampened stapedial mobility, thus increasing AC-sounds impedance, in turn resulting in a low-frequency 273 CHL”. How could the saccular dilatation happen when both perilymphatic and endolymphatic pressure increased?

4.     On page 8 L 281-284: “Accordingly, EH signs were detected in our patient through an extensive vestibular assessment: in particular, the dissociation between BCT and vHIT results and reduced VEMPs to AC and 283 BC-sounds addressed or first diagnosis toward MD [33–38]”. This is doubtful evidence.

Author Response

We thank the Editor of “Audiology Research” and all Reviewers for having considered our manuscript. We sincerely appreciate the valuable and detailed comments of the Reviewers and strongly believe the manuscript has been substantially improved by these suggestions.

A response to all your valuable comments is reported below (in bold), whereas the changes are highlighted in yellow in the revised copy of the manuscript.

We hope to have correctly corrected all unclear and wrong aspects throughout the text and hope that our revised manuscript may have addressed all comments.

Review 1

The manuscript titled with “Low-frequency Air-bone Gap and Pulsatile Tinnitus due to Dural Arteriovenous Fistula: Considerations upon Possible 3 Pathomechanisms and Literature Review” reported an interesting case of pulsatile tinnitus due to dural arteriovenous fistula. The authors discussed on the possible mechanism. It is the second case of the similar manifestation in the literature. It is relevant to the clinic to report the second case. However, the manuscript needs revision.

  1. The sentence on page 7 L227-228 that “Finally, Cassandro et al. reported a complete receding of CHL and PT after an endovascular embolization of a DAVF, similarly to our patient” should gone to introduction instead of discussion.

Response: The Reviewer is totally right. Being the first and unique description of a similar case, quoting the paper of Cassandro et al. is of pivotal importance and it has been moved to the introduction of the paper in lines 123-25: ”A similar case has previously been described by Cassandro et al. and our case represents the second description of this unusual combination in the literature [39]

  1. On page 8 L263-264: “This mechanism might lead to non-structural and reversible clinical manifestation even far from the AVM nidus, generating a subclinical intracranial hypertension, as reported by Hurst et al. [55]”. Did the authors measure CSF pressure in the present case?

Response: The question of the Reviewer is more than fair. Unfortunately, we did have not the chance to measure CSF pressure. We specified this shortcoming in the conclusion of the paper in line 305-306: “Neither CSF pressure was measured nor the presence of papilledema was checked.”

  1. On page 8 L271-273: “The increased perilymphatic and endolymphatic pressure, together with a possible saccular dilatation, could likely result in a dampened stapedial mobility, thus increasing AC-sounds impedance, in turn resulting in a low-frequency 273 CHL”. How could the saccular dilatation happen when both perilymphatic and endolymphatic pressure increased?

Response: This assumption only represents one of the possible speculations concerning the putative pathomechanisms accounting for conductive hearing loss in case of dural arteriovenous fistula, hypothesizing a hydropic dilation of the membranous labyrinth. Other authors hypothesized similar mechanisms to explain the onset of conductive hearing loss in case of Meniere’s disease. In our thoughts, it should be included among possible mechanisms.

  1. On page 8 L 281-284: “Accordingly, EH signs were detected in our patient through an extensive vestibular assessment: in particular, the dissociation between BCT and vHIT results and reduced VEMPs to AC and BC-sounds addressed our first diagnosis toward MD [33–38]”. This is doubtful evidence.

Response: The doubts of the Reviewer are more than fair. Nevertheless, we did have not have other tests in our hands to objectively verify an endolymphatic hydrops in the patient herein described. We specified all these shortcomings at the end of the paper in lines 306-3012 “..delayed MRI acquisitions following contrast injection to objectively detect signs of EH were not obtained as 3T MRI was not available in our institution. Then, despite the patient herein reported exhibited a MD-like instrumental profile, she never reported acute vertigo spells neither fluctuating auditory symptoms, resulting in an incomplete clinical picture for MD diagnostic criteria [68]. Additionally, neither frequency tuning for VEMPs nor electrocochleography were performed to provide additional clues for EH.”

Reviewer 2 Report

Comments and Suggestions for Authors

RTEVIIEW OF LOW -FREQUENCY AIR BONE GAP AND PULSATILE TINNITUS DUE TO DURAL ARTERIOVENOUS FISTULA: Considerations upon possible Pathomechanisms and Literature Review

This is an interesting case, well written describing a patient with low frequency air/bone gap and objective pulsatile tinnitus and normal impedance audiometry that mimicked a third window syndrome. The patient reported headache and dizziness, detailed evaluation showed a mid-caloric canal paresis, and conductive low -frequency hearing loss, with normal impedance. The video HIT and VNG were normal. The patient had impaired c and o VEMP”s.

A head MRI showed a dural AV fistula (DAVF) between the left occipital artery and the transverse sinus, confirmed later by angiography.

The list of entities causing pulsatile tinnitus is comprehensive. In this case unlike the patient with IIH, there was an audible bruit over the left mastoid tip and was abolished by compression of the mastoid. This finding would not be present in IIH. Unlike other third window syndrome this objective bruit was extremely suggestive of a vascular cause, and DAVF would be on the top of the differential. The temporal bone scan excluded canal dehiscence or jugular bulb abnormality and MRA and angiography confirmed the DAVF.

There are several minor points that require comment:

1.What was the patient’s BMI: If long standing DVAF can cause increased Icp. In such cases one might find papilledema. What was the optic disc like?

2.What was her BMI and did she have an empty sella, or distention of the perioptic subarachnoid sheath.

Author Response

We thank the Editor of “Audiology Research” and all Reviewers for having considered our manuscript. We sincerely appreciate the valuable and detailed comments of the Reviewers and strongly believe the manuscript has been substantially improved by these suggestions.

A response to all your valuable comments is reported below (in bold), whereas the changes are highlighted in yellow in the revised copy of the manuscript.

We hope to have correctly corrected all unclear and wrong aspects throughout the text and hope that our revised manuscript may have addressed all comments.

Review 2

REVIEW OF LOW -FREQUENCY AIR BONE GAP AND PULSATILE TINNITUS DUE TO DURAL ARTERIOVENOUS FISTULA: Considerations upon possible Pathomechanisms and Literature Review

This is an interesting case, well written describing a patient with low frequency air/bone gap and objective pulsatile tinnitus and normal impedance audiometry that mimicked a third window syndrome. The patient reported headache and dizziness, detailed evaluation showed a mid-caloric canal paresis, and conductive low -frequency hearing loss, with normal impedance. The video HIT and VNG were normal. The patient had impaired c and o VEMPs. A head MRI showed a dural AV fistula (DAVF) between the left occipital artery and the transverse sinus, confirmed later by angiography. The list of entities causing pulsatile tinnitus is comprehensive. In this case unlike the patient with IIH, there was an audible bruit over the left mastoid tip and was abolished by compression of the mastoid. This finding would not be present in IIH. Unlike other third window syndrome this objective bruit was extremely suggestive of a vascular cause, and DAVF would be on the top of the differential. The temporal bone scan excluded canal dehiscence or jugular bulb abnormality and MRA and angiography confirmed the DAVF.

Response: we thank the Reviewer for his/her kind comments on our case report.

There are several minor points that require comment:

1.What was the patient’s BMI: If long standing DVAF can cause increased Icp. In such cases one might find papilledema. What was the optic disc like?

Response: Reviewer’s comments are absolutely right. BMI of the patient was 21 and we included this finding in the case presentation in line 137. “Her body mass index was 21”.

Unfortunately, the presence of papilledema was not checked and we included this shortcoming among the limitation of the paper in lines 305-306. “Neither CSF pressure was measured nor the presence of papilledema was checked”.

2.What was her BMI and did she have an empty sella, or distention of the perioptic subarachnoid sheath.

Response: Reviewer’s comments are absolutely right. MRI excluded signs of empy sella and distention of the perioptic subarachnoid sheath. We included these details in the case description in lines 155-56. “Neither signs of empty sella nor distention of the perioptic subarachnoid sheaths was found.”